# "Survey on the burden, epidemiological and clinical characteristics of snakebite envenoming in the Health Demographic Surveillance System (HDSS) of Taabo (Southern Côte d'Ivoire)"

**Tanoh Amany Serge Raymond N'Krumah**[1,2]*, **Bognan Valentin Koné**[2], **Yao Didier Koffi**[2,3], **Dognimin Ismael Coulibaly**[1,2], **Abdoulaye Tall**[2], **Siaka Koné**[2], **Simone Toppino**[4], **Marija Stojkovic**[4], **Bassirou Bonfoh**[2], **Thomas Junghanss**[4]

**1** Unité de Formation et de Recherche des Sciences Biologiques, Université Peleforo Gon Coulibaly, Korhogo, Côte d'Ivoire, **2** Département Recherche et Développement, Centre Suisse de Recherches Scientifiques en Côte d'Ivoire, Abidjan, Côte d'Ivoire, **3** Programme National de Lutte contre l'Ulcère de Buruli et des Maladies Cutanées Ulcérative, Abidjan, Côte d'Ivoire, **4** Division Infectious Diseases and Tropical Medicine, Heidelberg University Hospital, Heidelberg, Germany

* nktanoh@gmail.com

## Abstract

### Background

In Sub-Saharan Africa, reliable data on the burden, clinical epidemiology and management of snakebites remains scarce. Mostly, crude population and institution-based estimates have been reported. The aim of this study was to estimate the burden and describe the epidemiological and clinical characteristics of snakebite envenoming based on population- (HDSS) data.

### Methods

A cross-sectional snakebite survey was conducted in all households in the Ahondo Health Area (AHA), a sub-population of the Taabo Health and Demographic Surveillance System (HDSS) between October 2 to December 22, 2023. All household members were interviewed, and a questionnaire was administered to snakebite cases. Epidemiological and clinical characteristics were described, and the annual snakebite incidence calculated.

### Results

795 households were surveyed, representing 3,924 people in the AHA. The annual incidence of snakebites per 100,000 population was 280.3 (CI: 140.0 - 501.0), the case-fatality rate 2.0% (95%CI: 0.04 - 10.5) and the median age of snakebite victims 43 years. The annual incidence of snakebites was significantly higher in the seasonally used agricultural camps (3296.7 cases per 100,000 inhabitants) than in the villages (208.7 cases per 100,000 inhabitants). Snakebites most frequently occurred in the morning and at night.

**Data availability statement:** All relevant data are included in the manuscript and its supporting information files.

**Funding:** This work was supported by the Else Kröner-Fresenius-Stiftung and Swiss Tropical and Public Health Institute (SwissTPH) under Proj. No/Partner: 2021_EKHA.091 / Uni Heidelberg / Swiss TPH. The funders had no role in study design, data collection and analysis, decision to publish, or preparation of the manuscript.

**Competing interests:** The authors have declared that no competing interests exist.

Patients employed dangerous "first aid" measures. Most snakebite victims were treated by traditional healers. None of the cases attending the health services received antivenoms.

## Conclusion

Snakebite envenoming remains a threat for rural communities at an unacceptable high level unchanged since the second half of the last century. Reliable community-based data on the incidence, circumstances and management of snakebite envenoming is only the first step to mitigate the neglected health problem. Implementation of prevention measures, first aid and clinical management, including access to antivenoms, must immediately follow.

### Author summary

Snakebite envenoming is an important health threat to rural communities in many parts of the world. In Sub-Saharan Africa reliable data on the burden, clinical epidemiology and management of snakebites remain scarce. In our wound management project "Prevent, identify & treat wounds early: An integrated sectoral community-based approach" we found a substantial proportion of wounds which were most probably caused by snakebites. They were neither linked to snakebite by patients nor diagnosed as snakebite wounds by health staff in patients presenting to the health services. This observation prompted us to add an in-depth snakebite study to our wound management project estimating the burden and describing the epidemiological and clinical characteristics of snakebites. The annual incidence of snakebites in our study area was 280 cases per 100,000 population unchanged since the 70ies of the last century. The death of a 13-year-old teenager bitten by a green mamba is a very sad avoidable death due to the lack of education in snakebite prevention and first aid at the community and training and resources including antivenom at the health services level.

## Introduction

Snakebite envenoming listed as a Neglected Tropical Disease (NTD) in 2017 is a disease of the rural poor. Every year, it causes worldwide a substantial number of deaths and permanent disability in survivors, including restricted mobility, amputation, blindness, and psychological effects [1–6]. Snakebite and snakebite envenoming is reviewed by Gutiérrez et al [7], Warrell [8], and for Africa by Chippaux [9]. The WHO Snakebite information and data platform is a continuously updated resource [10].

In Sub-Saharan Africa, reliable data on the burden, prevention and management of snakebites remains scarce. Mostly, crude population and institution-based estimates have been reported [9,11,12]. The latter are biased since most patients do not reach health facilities and, additionally, are not systematically reported.

In Sub-Saharan Africa, the annual incidence estimates are 56 per 100,000 population (95% CI: 45–68/100 000) based on hospital data and 204 per 100,000 population (95% CI: 172–237/100 000) based on household surveys with a case fatality rate between 2.8% and 11.6% depending on whether antivenom was used. The incidence of sequelae may be in the range of 5% and amputations 3% [12].

Snake bite deaths are estimated at 20 000 – 32 000 people per year, most likely a cross underestimation [13].

High at-risk groups are rural agricultural workers, pastoralists, fishermen, hunters, children, people living in poorly constructed houses, and those with limited access to education and healthcare [9,14]. The shared environment of humans and venomous snakes and its implications has been described [9]; very detailed for the most important venomous snake in West Africa the saw scaled viper (*Echis carinatus*) [15].

The primary prevention measures, supportive and specific (antivenom) treatment are poorly developed across sub-Saharan Africa [9]. Guidelines for the prevention and management of snakebite envenoming have been published by WHO in 2010 [16].

In West Africa at least 3500–5350 deaths happen annually, equivalent to 1.2 deaths/100 000 population per annum (95% CI: 0.9–1.4/100 000) [17]. The underestimation of the snakebite burden in West Africa has been assessed by Habib et al [17].

In Côte d'Ivoire, a national household and health center-based epidemiological snakebite survey was carried out in 1979 and published in 2002, estimating the annual incidence of snakebites in rural areas at greater than 200 bites per 100,000 inhabitants, higher in forest areas (195 per 100,000 inhabitants) than in savannah areas (130 per 100,000 inhabitants).

The fatality rate was higher in savannahs (3.1%) than in forests (2.0%). More than half the bites occurred in men aged between 15 and 50, with a significantly higher risk among farmers [18]. Tiassalé health district, to which Taabo Health Demographic and Surveillance System (HDSS) belongs, was leading the snakebite case fatality rate (8.25%) in the national survey [18].

The Ahondo Health Area where our wound management project "Prevent, identify & treat wounds early: An integrated sectoral community-based approach" is carried out since 2019, is located in Taabo HDSS [19,20]. We found a substantial proportion of wounds which were most probably caused by snakebites. They were neither linked to snakebite by patients nor diagnosed as snakebite wounds by health staff in patients presenting to the health services. This observation prompted us to add an in-depth snakebite study to our wound management project.

The objective of the study was to estimate the annual snakebite incidence and to describe the epidemiological and clinical characteristics of snakebites in the HDSS setting of Ahondo Health Area (AHA).

## Methods

### Ethics statement

This study is part of the humanitarian and research project entitled "Prevent, identify & treat wounds early: An integrated sectoral community-based approach" carried out in the Taabo Health Demographic Surveillance System (HDSS). The study protocol was approved by the National Health and Life Sciences Ethics Committee (CNESVS) of Côte d'Ivoire (N/Ref: 025–22/MSHPCMU/CNESVS-km, dated March 03, 2022) and by the Ethical Review Board of the University Hospital of Heidelberg, Germany (N˚ 2021_EKHA.91, dated September 02, 2021). Written informed consent was obtained from all participants aged 18 years and older, or from parents, or legal representatives of persons younger than 18 years.

### Study area

The study took place in the Ahondo Health Area (AHA) including Ahondo and Sahoua villages. AHA is part of the Taabo Health Demographic Surveillance System (HDSS) located in the Tiassalé health district (Fig 1).

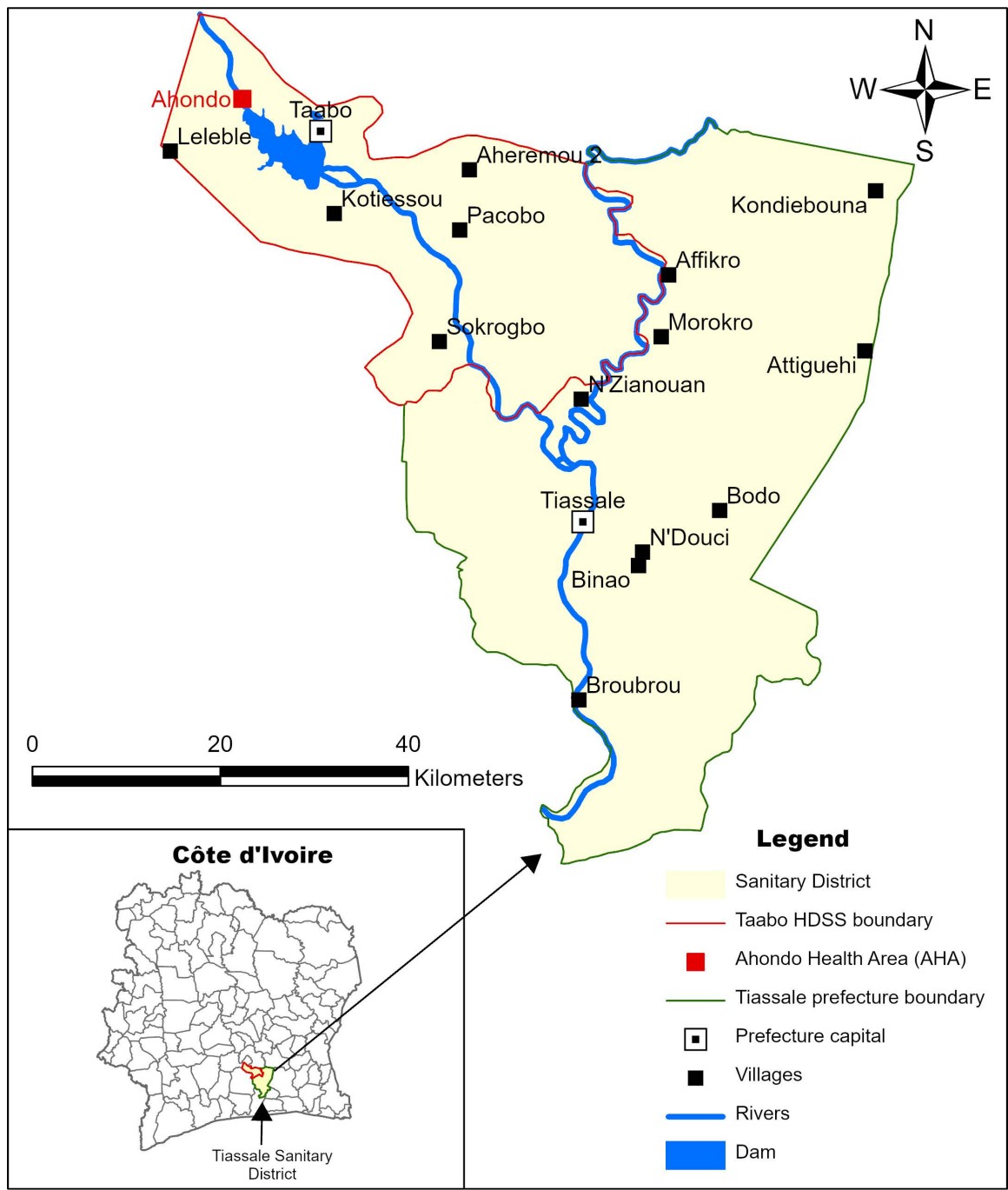

**Fig 1. Map of the Tiassalé Sanitary District in southern Côte d'Ivoire, including Taabo HDSS and Ahondo Health Area (Source: N'Krumah et al, [ 26], https://doi.org/10.1371/journal.pntd.0004327.g001).**

The Taabo HDSS is a research station of the Centre Suisse de Recherche Scientifique en Côte d'Ivoire (CSRS) located in south-central Côte d'Ivoire, approximately 150 km northwest of Abidjan. Since its creation in 2009, research on integrated control of water-related diseases, such as schistosomiasis, malaria, and Buruli ulcer, etc., have been carried out. The Taabo HDSS covers an area of approximately 980 km$^2$ located between latitudes 6°0' and 6°20' N and

longitudes 4°55' and 5°15' W. The area is mainly rural and comprises 13 villages as well as an urban settlement (Taabo-Cité) [21].

The annual rainfall ranges between 600 and 2000 mm per year, with four distinct climatic seasons [22]. In Taabo, a hydroelectric dam of about 69 km² was built on the Bandama River in the late 1970s. In 2019, AHA had a population of around 4,200 people representing 10% of the Taabo HDSS population. The livelihood mainly depends on agriculture. Ahondo Health Center (AHC) is the largest center in the health area and is located 15 km from the Taabo district hospital. Since December 2019, the Sahoua Health Center (SHC) has become active. The socio-sanitary situation of the Taabo HDSS in general and the AHA in particular is characterized by a high prevalence of wounds and Skin NTDs [19], communicable diseases such as malaria and schistosomiasis [23,24] and limited access to primary healthcare and safe drinking water [25]. In Ahondo, on the banks of the Bandama River, bananas are cultivated in plantations employing people of the villages. The banana plantation company is responsible for significant migrant worker movement in and out of AHA. The population of the AHA is diverse and representative of the Ivorian population composition.

## Study design, definition of a "snakebite" case and rationale for the presented groups

We conducted a cross-sectional exhaustive survey based on the entire population of the Ahondo health area which is part of the Taabo HDSS. There were no exclusion criteria.

We capitalized on the HDSS infrastructure where demographic, health, socioeconomic, environmental, etc. data are routinely collected to analyze snakebite envenoming in its context.

Definition of a 'snakebite' case and rational for the presented groups

Definition of a 'snakebite' case:

- A person reporting a snake/ the local name of the snake inflicting the bite.

- signs & symptoms interpreted as being caused by a snakebite based on the features described in the syndromic approach to snakebite envenoming in Sub-Saharan Africa [27,28].

- Formal identification of the offending snake by an expert familiar with the venomous snakes of the region.

Group 1: Persons in AHA reporting a snakebite as defined above between October 2, 2022 and October 2, 2023.

Assuming a low recall bias for snakebites (a dramatic event) within the previous 12 months, the annual snakebite incidence is calculated from Group 1 and the epidemiology and circumstances of the bites described.

Group 2: Persons in AHA reporting a snakebite as defined above before October 2, 2022.

The data of group 2 are used to get a broader insight into the regional snakebite problem accepting a higher recall bias compared to group 1.

## Data collection

The household survey was carried out in AHA between October 2 and December 22, 2023, by a team consisting of two interviewers from the Taabo HDSS, trained in data collection using the Open Data Kit (ODK) software, a Community Health Worker (CHW), a physician, a sociologist, and an epidemiologist.

The snakebite survey is part of an ongoing project to assess the wound burden, and the impact of the community-based wound management model implemented in 2019. Since the project had just started replacing paper questionnaires/ CRFSs by ODK, the data was collected by two interviewers with an ODK tablet and a paper-based questionnaire to capture and correct data entry errors. To minimize absences from households, the survey was carried out in the evenings between 4 p.m. and 8 p.m., after household members had returned from their daily activities.

Once in the household, the interviewers explained the purpose of the survey to the head of the household and asked each household member if they had been a victim of snakebites at any time in their live with particular emphasis on snakebites they experienced in the past twelve months prior to the household survey. People temporarily absent from the household during the survey were systematically revisited the next day by the investigators. In households, in which a victim of snake bite was identified, a specific questionnaire was administered directly to the person of 10 years old or older and to the parents if the victim was less than 10 years old and after written informed consent of the victim had been obtained from adults (≥18 years old) or the parent/legal guardian if the victim was a minor (<18 years old). For the only fatal case, a teenager who died following a snakebite, the questionnaire was administered to his mother using the verbal autopsy method [29,30].

Data collected on snakebite victims included socio-demographic characteristics, environmental and social risk factors, and clinical characteristics (Table 1).

In addition, an interview was conducted with two nurses of AHA, a doctor of Taabo General Hospital and the manager of Taabo pharmacy. These interviews focused on

- the medical treatment of snakebite;

- the use of antivenom;

- the availability of antivenom in Taabo pharmacy.

### Data management and analysis

Data was entered and uploaded using Open Data Source (ODK central v2023) and exported to a Microsoft Excel CSV file. R software (version 4.3.1) was used for data analysis. The annual incidence of reported snakebites per 100,000 population for the AHA population was calculated from snakebites reported for the period October 2, 2022, to October 2, 2023, of Group 1. The proportions of sociodemographic, clinical and environmental variables were calculated for both groups. Results of group 1 and 2 are separately tabulated. The Fisher test statistics was used to compare the proportions of categorical variables at the 5% threshold. The interviews were analyzed to identify the main elements of interest of our study.

## Results

### Demographic and epidemiological data of the snakebite survey

During the survey, 795 households were enumerated, 4,474 people were counted, and 3,924 people surveyed. 519 people had out-migrated, and 31 people had died. A total of 11 snakebite cases were reported in the AHA population for the period October 2, 2022, to October 2, 2023 (Group 1), and 51 people currently living in the AHA suffered snakebites before October 2, 2022 (Group 2). The annual snakebite incidence per 100,000 population calculated from group 1 was 280.3 per 100,000 (CI: 140.0 - 501.0). The AHA snakebite case-fatality rate calculated from group 2 was 2.0% (95% CI: 0.05 - 10.5) (Table 2).

**Table 1. Data collected from snakebite victims at the household survey in the Ahondo Health Area (AHA).**

| Socio-demographic and economic characteristics | Environmental characteristics | Snakebite characteristics |
|---|---|---|
| Household HDSS ID | Time of day of the bite | Snakebite within the past 12 months |
| Individual HDSS ID | Activity at the time of the bite | Snakebite prior to the past 12 months |
| Household size | Environment of the bite | |
| Place of residence | | Snake observed at/ immediately after the bite (escaping snake) |
| | | Snake observed and "local name" reported<br>Snake species identified by an expert |
| Sex | | Signs & symptoms interpreted as being caused by a snakebite |
| Age | | A health professional diagnosing a snakebite |
| Nationality | | |
| Area of residence | | |
| School level | | Location of the bite |
| Marital status | | |
| Profession | | Swelling at the bite site |
| | | Swelling of the arm/leg or more extended |
| Housing type | | Inability to raise the head |
| Nature of the household wall | | Inability to lift the eyelids |
| Nature of household soil | | Inability to lift arms and/or legs |
| Location of household members' defecation | | Bleeding from the bite site |
| Main water supply source | | Bleeding from the mouth |
| Main source of household lighting | | Red urine |
| Main household cooking method | | Loss of consciousness |
| Source of household income | | |
| | | Application of a black stone |
| | | Incision of the bite site<br>Application of a tourniquet |
| | | Self-medication<br>• medical products<br>• traditional products |
| | | Treatment<br>• health services<br>• traditional healer |
| | | Antivenom treatment at hospital |
| | | Victim living/ deceased |

HDSS: Health demography surveillance system.

**Table 2. Demographic and epidemiological data of snakebite victims of the household survey in the Ahondo Health Area (AHA).**

|  | Ahondo Health Area (AHA) |
|---|---|
| **Demographic data** |  |
| Number of households surveyed | 795 |
| Number of individuals surveyed in households | 3924 |
| Number of people out-migrated | 519 |
| Number of deceased individuals | 31 |
| **Epidemiological data** |  |
| Total cases | 62 |
| **Group 1 (Victims of snakebite between October 2, 2022, and October 2, 2023)** |  |
| Number of cases | 11 |
| Deaths attributed to snakebite | 00 |
| Annual incidence of snakebite per 100,000 population (95%CI) | 280.3 (140.0 – 501.0) |
| **Group 2 (Victims of snakebite before October 2, 2022)** |  |
| Number of cases | 51 |
| Deaths attributed to snakebite | 01 |
| Case fatality rate as a percentage (95%CI) | 2.0 (0.05 – 10.5) |

CI: confidence interval.

## Socio-demographic characteristics of snakebite victims

The demographic characteristics of snakebite victims in AHA Group 1 are summarized in Table 3.

In AHA, the annual incidence of snakebites was significantly higher in the camps (3296.7 cases per 100,000 inhabitants) than in the villages (208.7 cases per 100,000 inhabitants). The median age of snakebite victims was 43 years. For variables such as sex, age, nationality, profession, marital status and level of education of snakebite victims, there was no statistically significant difference between the different incidence classes at the 5% threshold.

The characteristics of the households of group 1 snakebite victims are presented in Table 4. Except for the "type of wall" variable in the households of group 1 snakebite victims, there was no statistically significant difference between the incidence categories of the other variables in the characteristics of the households of group 1 snakebite victims at the 5% threshold.

## Reported clinical characteristics and socio-environmental risk factors

**Clinical characteristics of snakebite victims.** The clinical characteristics of the snakebite victims are presented in Table 5.

Most victims reported signs and symptoms of envenoming. The majority of cases the bite occurred at the lower limb.

Swelling and bleeding at the bite site were most commonly observed.

More than half the cases applied a tourniquet after the snakebite. Nearly a third of snakebite victims consulted a health center, while more than half of cases a traditional healer.

None of the patients received antivenom according to health staff: "*No snakebite victims treated in rural health centers receive antivenom, because we don't have any. Patients are treated with antibiotics and anti-inflammatories and given anti-tetanus serum*".

Victims attributing a local name to the snakebite experienced mentioned Poponé (Vipera sp.), Woblé (Naja sp.) and *Mamba vert* (Dendroaspis sp.) as culprits. None of the culprits have been formally identified by an expert.

**Table 3. Demographic characteristics of group 1 snakebite victims of the household survey in the Ahondo Health Area (AHA).**

| Variables | Group 1 (n = 11) | | | | |
|---|---|---|---|---|---|
| | Number (n) | Denominator (N) | Annual incidence per 100,000 | Confidence Interval (CI) | p-value |
| **Sex** | | | | | |
| Male | 9 | 2022 | 445.1 | (203.7 - 843.3) | 0.067 |
| Female | 2 | 1902 | 105.2 | (12.7 - 379.3) | |
| **Age in years: median (range)** | 43 (9-66) | | | | |
| < 15 | 1 | 1191 | 84.0 | (2.1 - 466.9) | 0.293 |
| 15 - 34 | 4 | 1212 | 330.0 | (90.0 - 842.8) | |
| 35 - 54 | 5 | 989 | 505.6 | (164.4 - 1175.8) | |
| ≥ 55 | 1 | 532 | 188.0 | (4.8 - 1042.8) | |
| **Nationality** | | | | | |
| Côte d'Ivoire | 6 | 2782 | 215.7 | (79.2 - 468.8) | 0.264 |
| Burkina Faso | 4 | 740 | 540.5 | (147.5 - 1378.2) | |
| Mali | 1 | 297 | 336.7 | (8.5 - 1861.6) | |
| **Area of residence** | | | | | |
| Village | 8 | 3833 | 208.7 | (90.1 - 410.8) | **0.002*** |
| Camps | 3 | 91 | 3296.7 | (685.1 - 9333.2) | |
| **School level** | | | | | |
| No school level | 7 | 1873 | 373.7 | (150.4 - 768.5) | 0.058 |
| Primary | 3 | 1266 | 237.0 | (48.9 - 690.9) | |
| Secondary | 1 | 785 | 127.4 | (3.2 - 707.7) | |
| **Marital status** | | | | | |
| Single/Divorced/Widowed | 5 | 1809 | 276.4 | (89.8 - 643.8) | 0.116 |
| Married (traditional, civil) | 5 | 924 | 541.1 | (175.9 - 1258.3) | |
| Not applicable | 1 | 1191 | 84.0 | (2.1 - 466.9) | |
| **Profession** | | | | | |
| Farmer | 7 | 1735 | 403.5 | (162.4 - 829.5) | 0.072 |
| Student | 1 | 630 | 158.7 | (4.0 - 881.2) | |
| Housewife | 2 | 104 | 1923.1 | (233.7 - 6774.5) | |
| Other professions | 1 | 699 | 143.1 | (3.6 - 794.5) | |

CI: confidence interval * Statistically significant.

**Socio-environmental findings of snakebites and snakes involved.** Socio-environmental findings reported by snakebite victims are presented in Table 6. Snakebites occurred most often during the main activity periods of the day and at night, most frequently in the fields during farm work.

## Discussion

The snakebite study conducted in the general population of the Ahondo Health Area (AHA) provides, for the first time, population (HDSS) based epidemiological and clinical characteristics of snakebites in southern Côte d'Ivoire. The snakebite survey is part of a research project assessing the wound burden and clinical epidemiology and evaluating the impact of a community-based wound management model implemented in 2019 and monitored since then [19,20]. The repeated wound surveys cover the entire AHA population.

We divided the data collected into two groups. To minimize recall bias for the calculation of the annual snakebite incidence, we used the data of the last 12 months (October 2, 2022,

**Table 4. Household characteristics of group 1 snakebite victims of the household survey in the Ahondo Health Area (AHA).**

| Variables | Group 1 (n = 11) | | | | |
|---|---|---|---|---|---|
| | Number (n) | Denominator (N) | Percentage (%) | (95%CI) of Percentage | p-value |
| **Housing type** | | | | | |
| Traditional house | 1 | 312 | 0.3 | (0.01 - 1.8) | 0.058 |
| Traditional-modern house | 10 | 483 | 2.1 | (1.0 - 3.8) | |
| **Nature of the household wall** | | | | | |
| Cement brick wall | 7 | 203 | 3.4 | (1.4 - 7.0) | **0.010*** |
| Earth brick wall | 4 | 592 | 0.7 | (0.2 - 1.7) | |
| **Nature of household soil** | | | | | |
| Cemented | 11 | 771 | 1.4 | (0.8 - 2.5) | – |
| Uncemented | 0 | 24 | 0 | – | |
| **Location of household members' defecation** | | | | | |
| Latrine | 9 | 572 | 1.5 | (0.7 - 3.0) | 0.737 |
| Bush | 2 | 223 | 0.9 | (0.1 - 3.2) | |
| **Main water supply source** | | | | | |
| Village pump | 9 | 673 | 1.3 | (0.6 - 2.5) | 0.681 |
| Rivers | 2 | 122 | 1.6 | (0.2 - 5.8) | |
| **Main source of household lighting** | | | | | |
| Electricity | 10 | 693 | 1.4 | (0.7 - 2.6) | 0.999 |
| Other light source | 1 | 102 | 1 | (0.02 - 5.3) | |
| **Main household cooking method** | | | | | |
| Firewood | 8 | 541 | 1.5 | (0.6 - 2.9) | 0.999 |
| Others (Gas, charcoal) | 3 | 254 | 1.2 | (0.2 - 3.4) | |
| **Source of household income** | | | | | |
| Agricultural activities | 10 | 570 | 1.8 | 0.8 – 3.2 | 0.308 |
| Other activities | 1 | 225 | 0.4 | 0.01 2.5 | |

CI: confidence interval * Statistically significant.

and October 2, 2023) prior to the time when the survey was conducted (group 1). For all other variables we merged the two data sets. On comparison, group 1 and group 2 data were interestingly very consistent.

The annual incidence of snakebites in AHA calculated from the group 1 data was 280.3 (CI: 140.0 - 501.0) per 100,000 inhabitants, exceeding the estimate for rural Côte d'Ivoire based on data of the 1970ies with 200 bites per 100,000 inhabitants [18].

In the AHA, the incidence of snakebites was significantly higher in the "camps" - agricultural areas where people periodically live and do seasonal field work - than in and around villages.

80% of snakebite victims identified in the 12 months preceding the survey (group 1), were aged between 15 and 54, an age range which was also reported by others [18,31,32].

The low incidence of snakebite victims aged under 15 years could be explained by the fact that children in this age range are under the close care of adults or are attending school and are therefore not overly involved in the agricultural working activities that are the main cause of snakebites.

The snakebite fatality rate in AHA group 2 was 2.0%. The snakebite case fatality rate found in our study corresponds to previous estimates of 2% in the forest areas, where the Taabo HDSS is located, and 3% in the savannah zone [18].

**Table 5. Characteristics of the snakebites reported by victims of the household survey in the Ahondo Health Area (AHA).**

| Variables | Group 1 (n = 11) | | | Group 2 (n = 51) | | |
|---|---|---|---|---|---|---|
| | Number (n) | Percentage (%) | (95% CI) of percentage | Number (n) | Percentage (%) | (95% CI) of percentage |
| **Location of the bite** | | | | | | |
| Head | 0 | 0 | (0 - 28.5) | 1 | 2.0 | (0.1 - 10.4) |
| Upper limbs | 1 | 9.1 | (0.2 - 41.3) | 8 | 15.7 | (7.0 - 28.6) |
| Lower limbs | 10 | 90.9 | (58.7 - 99.8) | 42 | 82.3 | (69.1 - 91.6) |
| **Swelling at the bite site** | | | | | | |
| Yes | 8 | 72.7 | (39.0 - 94.0) | 37 | 72.5 | (58.3 - 84.1) |
| No | 3 | 27.3 | (6.0 - 61.0) | 14 | 27.5 | (15.9 - 41.7) |
| **Swelling of the arm/leg and more extended** | | | | | | |
| Yes | 4 | 36.4 | (10.9 - 69.2) | 15 | 29.4 | (17.5 - 43.8) |
| No | 7 | 63.6 | (30.8 - 89.1) | 36 | 70.6 | (56.2 - 82.5) |
| **Inability to raise the head** | | | | | | |
| Yes | 0 | 0 | (0 - 28.5) | 1 | 2 | (0.1 - 10.4) |
| No | 11 | 100 | (71.5 - 100) | 50 | 98 | (89.6 - 99.9) |
| **Inability to lift the eyelids** | | | | | | |
| Yes | 0 | 0 | (0 - 28.5) | 1 | 2 | (0.1 - 10.4) |
| No | 11 | 100 | (71.5 - 100) | 50 | 98 | (89.6 - 99.9) |
| **Inability to lift arms and/or legs** | | | | | | |
| Yes | 0 | 0 | (0 - 28.5) | 1 | 2 | (0.1 - 10.4) |
| No | 11 | 100 | (71.5 - 100) | 50 | 98 | (89.6 - 99.9) |
| **Bleeding from the bite site** | | | | | | |
| Yes | 8 | 72.7 | (39.0 - 94.0) | 43 | 84.3 | (71.4 - 93.0) |
| No | 3 | 27.3 | (6.0 - 61.0) | 8 | 15.7 | (7.0 - 28.6) |
| **Bleeding from the mouth** | | | | | | |
| Yes | 0 | 0 | (0 - 28.5) | 0 | 0 | (0.0 - 7.0) |
| No | 11 | 100 | (71.5 - 100) | 51 | 100 | (0.93 - 100) |
| **Red urine** | | | | | | |
| Yes | 0 | 0 | (0 - 28.5) | 0 | 0 | (0.0 - 7.0) |
| No | 11 | 100 | (71.5 - 100) | 51 | 100 | (0.93 - 100) |
| **Loss of consciousness** | | | | | | |
| Yes | 0 | 0 | (0 - 28.5) | 1 | 2 | (0.1 - 10.4) |
| No | 11 | 100 | (71.5 - 100) | 50 | 98 | (89.6 - 99.9) |
| **Application of a black stone** | | | | | | |
| Yes | 3 | 27.3 | (6.0 - 61.0) | 16 | 31.4 | (19.1 - 45.9) |
| No | 8 | 72.7 | (39.0 - 94.0) | 35 | 68.6 | (54.1 - 80.9) |
| **Incision of the bite site** | | | | | | |
| Yes | 9 | 81.8 | (48.2 - 97.7) | 43 | 84.3 | (71.4 - 93.0) |
| No | 2 | 18.2 | (2.3 - 51.8) | 8 | 15.7 | (7.0 - 28.6) |
| **Self-medication** | | | | | | |
| Medical products | 3 | 27.3 | (6.0 - 61.0) | 9 | 17.6 | (8.4 - 30.9) |
| Traditional products | 8 | 72.7 | (39.0 - 94.0) | 42 | 82.4 | (69.1 - 91.6) |
| **Treatment** | | | | | | |
| Health services | 4 | 36.4 | (10.9 - 69.2) | 16 | 31.4 | (19.1 - 45.9) |
| Traditional healer | 7 | 63.6 | (30.8 - 89.1) | 35 | 68.6 | (54.1 - 80.9) |
| **Antivenom treatment at hospital** | | | | | | |
| Yes | 0 | 0 | (0.0 - 60.2) | 0 | 0 | (0.0 - 20.1) |
| No | 4 | 100 | (39.8 - 100) | 16 | 100 | (79.4 - 100) |

*(Continued)*

**Table 5.** (Continued)

| Variables | Group 1 (n = 11) | | | Group 2 (n = 51) | | |
|---|---|---|---|---|---|---|
| | Number (n) | Percentage (%) | (95% CI) of percentage | Number (n) | Percentage (%) | (95% CI) of percentage |
| **Applying a tourniquet** | | | | | | |
| Yes | 7 | 63.6 | (30.8 - 89.1) | 27 | 52.9 | (38.5 - 67.1) |
| No | 4 | 36.4 | (10.9 - 69.2) | 24 | 47.1 | (32.9 - 61.5) |
| **Snake observed at/ after the bite** | | | | | | |
| Yes | 8 | 72.7 | (39.0 – 94.0) | 31 | 60.8 | (46.1 – 74.2) |
| No | 3 | 27.3 | (6.0 – 61.0) | 20 | 39.2 | (25.8 – 53.9) |
| **Snake observed and attributed a "local name"** | | | | | | |
| Poponé (Vipera sp.) | 3 | 37.5 | (8.5 - 75.5) | 8 | 25.8 | (11.9 - 44.6) |
| Woblé (Naja sp.) | 2 | 25.0 | (31.9 - 65.1) | 7 | 22.6 | (9.6 - 41.1) |
| *Mamba vert* (Dendroaspis sp.) | 2 | 25.0 | (31.9 - 65.1) | 12 | 38.7 | (21.8 - 57.8) |
| No name reported | 1 | 12.5 | (0.3 - 52.7) | 4 | 9.7 | (2.0 - 25.7) |

CI: confidence interval.

**Table 6. Socio-environmental findings reported by snakebite victims of the household survey in the Ahondo Health Area (AHA).**

| Variables | Group 1 (n = 11) | | | Group 2 (n = 51) | | |
|---|---|---|---|---|---|---|
| | Number (n) | Percentage (%) | (95% CI) of percentage | Number (n) | Percentage (%) | (95% CI) of percentage |
| **Time of day of the bite** | | | | | | |
| Dawn of the day | 0 | 0 | (0.2 - 41.3) | 2 | 3.9 | (0.5 - 13.5) |
| Morning | 4 | 36.4 | (10.9 - 69.2) | 21 | 41.2 | (27.6 - 55.8) |
| Midday/Afternoon | 4 | 36.4 | (10.9 - 69.2) | 4 | 7.8 | (2.2 - 18.9) |
| Evening | 0 | 0 | (0.2 - 41.3) | 9 | 17.6 | (8.4 - 30.9) |
| Night | 3 | 27.3 | (6.0 - 61.0) | 15 | 29.4 | (17.5 - 43.8) |
| **Activity at the time of the bite** | | | | | | |
| Working in the farm | 7 | 63.6 | (30.8 - 89.1) | 20 | 39.2 | (25.8 - 53.9) |
| Walking/hiking | 0 | 0 | (0.2 - 41.3) | 5 | 9.8 | (3.2 - 21.4) |
| Household activities | 4 | 36.4 | (10.9 - 69.2) | 18 | 35.3 | (22.4 - 49.9) |
| Fishing activities | 0 | 0 | (0.2 - 41.3) | 6 | 11.8 | (4.4 - 23.9) |
| Leisure activities | 0 | 0 | (0.2 - 41.3) | 2 | 3.9 | (0.5 - 13.5) |
| **Environment of the bite** | | | | | | |
| At home | 4 | 36.4 | (10.9 - 69.2) | 15 | 29.4 | (17.5 - 43.8) |
| At the side of the road | 0 | 0 | (0.2 - 41.3) | 4 | 7.8 | (2.2 - 18.9) |
| In the field/in the forest | 7 | 63.6 | (30.8 - 89.1) | 25 | 49 | (34.8 - 63.4) |
| In the river | 0 | 0 | (0.2 - 41.3) | 6 | 11.8 | (4.4 - 23.9) |
| On a tree | 0 | 0 | (0.2 - 41.3) | 1 | 2 | (0.1 - 10.4) |

CI: confidence interval.

The snakebite death recorded in our study concerns a 13-year-old boy of the village Ahondo, He had been bitten several times on the head by a snake after climbing a coconut tree. "*Coming down from the coconut tree and shouting that he had been bitten by a green snake, this child died after a few hours at the village health center*", according to the parents of the deceased. The teenager's death can be attributed to a green mamba (Dendroaspis sp.), an elapid that has well adapted to humanized environments [33]. The very fast snake, with its

powerful neurotoxic venom, is a major cause of fatal envenoming in Sub-Saharan Africa. The death of this teenager demonstrates two problems at the health services level: (i) the lack of knowledge on immediate supportive treatment such as ventilation of the patient and, (ii) the lack of access to antivenom.

Incidence figures, the affected population and the physical environment where snakebites were reported confirmed that envenoming is clustered in rural hotspots and in specific segments of the population [34].

Snakes are mostly not or only vaguely seen when the bite occurs. Surprisingly, between 60 and 70% of the snakebite victims of our study claimed that they had observed the snake. Consistent between group 1 and 2, the proportion of the offending snakes (local names) were attributed in similar proportions to "Poponé" (Vipera sp.), "Woblé" (Naja sp.) and "*Mamba vert*", (Dendroaspis sp.) as the main culprits [35]. Snakebites occurred mainly during working hours (70%) and at night (27%), in farms and at home. The high proportion of snakebites among housewives (39%) could be explained by the fact that they are the first to get up in the morning for daily household activities, work in the gardens and in the fields. The risk of snakebites in and around the house at night, when snakes are most active, is high due to lack of lighting.

Bites occurred mainly at the lower extremities (90%) which is expected.

The clinical signs reported by most snakebite victims were swelling at the bite site (70%) and bleeding from the bite site (around 75%). In most cases reported clinical signs resolved without specific treatment leading to the wrong conclusion that the measures applied suffice in envenomed patients. The resolution indicates, instead, that only a small quantity of venom was injected if it was a venomous snake or the clinical signs observed were wrongly attributed to a venomous snake. The latter is a well-known reporting bias in incidents where a snake has not been unequivocally observed. This is especially a problem at night or in thick undergrowth away from footpaths. Dry bites which are in the range of 50% of venomous snake bites are additionally contributing to misinterpreting the danger on venomous snakes.

The fact that signs and symptoms of envenoming are often neither attributed to a snakebite by the victim nor, initially, by health staff has been confirmed in our wound management study.

Between May 2019 and July 2024, the wounds of thirteen out of more than three hundred patients with complicated or severe wounds treated at Taabo District Hospital were attributed to snakebite envenoming by the medical team of our wound management project (see Fig 2).

The evolution and the time course of large soft tissue necrosis around cytotoxic snakebites distinguishes them clearly from other causes. One of the snakes responsible for predominately local tissue damage in the study area are spitting cobras (Naja sp.) [36–38]

In a case series from Nigeria all 14 patients bitten by the spitting cobra *Naja nigricolis* reported developed swelling at the bite noticed after an interval of between three minutes and three hours. In most patients, the swelling was extended reaching its maximum within 36 hours of the bite. Transient local bleeding from the bite was noticed in five patients. Local tissue necrosis was detected in 10 patients within 5 days after the bite and complete breakdown followed quickly [36].

First aid is poorly established in Cote d'Ivoire. Black stones were used in 31% of snakebite victims in our study. Black stones are ineffective and there is very strong advice against their use [27]. Alarmingly 82% of the snakebite victims incised the skin at the bite side. Also, dangerous – particularly in regions with venomous snakes causing clotting disturbances - and, entirely ineffective. Between 50 and 60% applied tourniquets. Again here, a problem with first aid measures. If applied too tightly they cause significant local damage, if applied only loosely, they are useless.

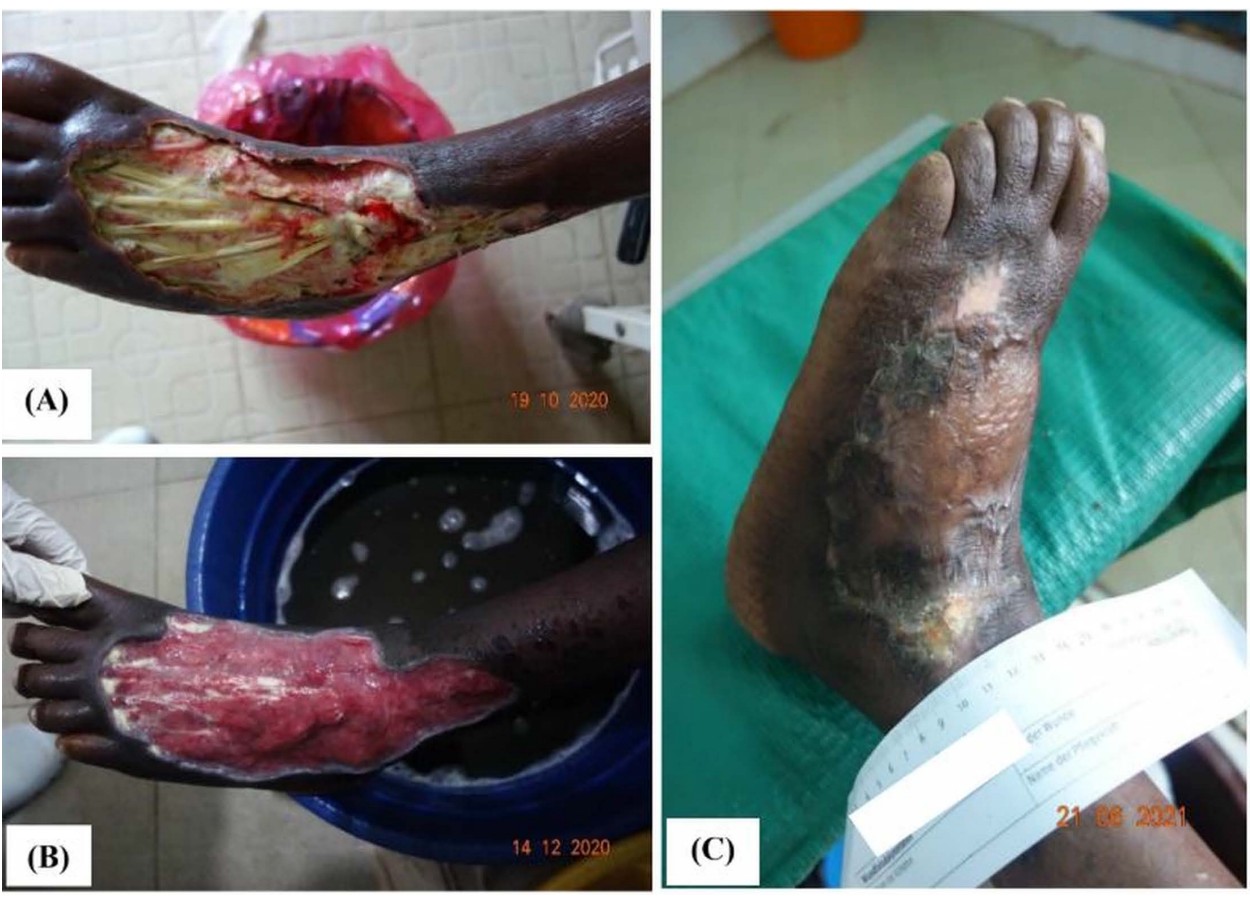

**Fig 2. 65-year-old man with a severe chronic wound on the forefoot and lower left leg presenting to the wound management unit of our project after 3 months of traditional treatment** (A). The patient was not aware of the snakebite and the health services he had attended did not suspect a snakebite as the cause. The course of events the patient reported was highly suggestive of a snakebite, most likely a spitting cobra (Naja sp.). Within 2 months of wound care healthy granulation tissue had developed (B). Final result after skin grafting (C).

In our study more than 60% of snakebite victims primarily consulted traditional healers. The high involvement of traditional healers in snakebite patients is one of the reasons why snakebite victims present late to the health services or not at all. At the same time, it is an opportunity if traditional healers could be convinced to cooperate and to be trained in snakebite prevention and professional referral to Health Centers and hospitals [39–41].

A snakebite management plan, equipment and materials, including very importantly antivenoms, are, however, preconditions for the cooperation and the development of trust in the health services. In our study none of the snakebite victims treated at Ahondo Health Center and Taabo District Hospital received antivenom. Most snakebite victims attending Ahondo Health Center and Taabo District Hospital were treated with antibiotics, anti-inflammatory drugs, and received tetanus prophylaxis. Interviewed health staff commented: "*This protocol for treating snakebite victims was put in place due to the lack of antivenom*". "*High cost, low demand, the expiry date of antivenom and power cuts in Taabo are the main reasons for not ordering antivenom*".

## Study limitations

The main limitation of the first snakebite study within the framework of our longitudinal population (HDSS) based wound management project is the size of the study population of around 4,000 inhabitants surveyed. Considering the size of the population it is remarkable that

the main features of snakebite envenoming (clinical, environmental and persisting managerial deficits) of the region are captured, and the annual incidence found is within the range of previous estimates. Despite the small population we survey, we could clearly demonstrate in this first snakebite study in Taabo HDSS that snakebite is a problem in the Taabo HDSS area and that it is unattended. The wound management project is close to rollout in a larger population the entire Taabo HDSS with 78.000 inhabitants which will present an opportunity for a more representative population sample with the methodology well established in the present study.

Estimating the incidence of snakebites and capturing the data on the circumstances of the bite, the culprit, the clinical presentation and course of events including first aid and treatment are hampered by a range of problems:

(a) snakebite vs snakebite envenoming: Counting snakebites is illusive; most snakebites go unnoticed. Studies on snakebites mainly capture snakebite envenoming. Patterns of signs and symptoms depending on the venom composition of the offending snake help to diagnose patients who experienced clinically relevant venom injections. Regionally, the envenoming pattern can be fairly reliably attributed to snake genus [27,42]. (b) recall bias: We minimized the recall bias, in particular to not underestimate snakebite (snakebite envenoming) incidence, by only using data of the past 12 months prior to the survey. Snakebite envenoming if substantial is traumatic and remembered even for longer periods of time. (c) observation bias: We cannot rely on patients attributing the right cause to the accident they experienced. This happens in both directions. Fast developing necrotic wounds as described above is an example of not attributing a snakebite to the clinical findings. To the contrary, snakebites are regularly blamed for a wound which in fact had been caused by a thorn etc. the patient stepped in. In our study also staff of health services did not attribute snakebites to wounds which most likely had been caused by envenoming.

In summary, the study results reveal information and figures on snakebite envenoming and not snakebites in general. Inquiring a short period of time (12 months) in an HDSS setting as in our study is as close as possible to robust estimates of snakebite envenoming incidence and outcomes. An underestimation is likely due to a bias towards more severe envenoming and due to an observation bias attributing the clinical finding to other causes. This may be partly balanced by an overestimation due to an observation bias attributing other causes of the clinical findings to a snakebite.

Despite limitations, the results of this first study of snakebites in the Taabo HDSS are valuable to guide prevention, first aid and clinical management strategies and informing future research, as widely recommended [3,9,45].

In conclusion, the results of our study show that the snakebite burden in rural Southern Côte d'Ivoire remains in the range of what had been recorded in the second half of the last century and continues to be a significant health threat to the rural population at work and at home. First aid measures and clinical diagnosis and management including antivenom availability in the health services are entirely inappropriate. The dominant role of traditional healers cannot be converted into cooperation under these circumstances. A countrywide effort is urgently required to mitigate the threat of snakebite envenoming. The main points are (a) creating awareness and educating communities in snakebite prevention, first aid and recognition as an emergency requiring immediate referral to the health services, (b) training staff of health services in diagnosing and managing snakebite envenoming, (c) providing life-saving equipment (e.g., to manage respiratory failure), (d) stocking antivenoms and instruct its application (d) securing follow-up and rehabilitation including mental problems.

## Supporting information

**S1 Checklist. STROBE Checklist.** Statement—checklist of items that should be included in reports of observational studies.
(PDF)

**S1 File. Survey form on snakebite envenoming in the Ahondo health area (Taabo HDSS - Côte d'Ivoire).**
(PDF)

## Acknowledgments

We would like to thank the investigators of the Health and Demographic Surveillance System (HDSS) of Taabo and the community health workers for their active involvement during the data collection in the Ahondo Health Area, the community members, the staff of the Ahondo and Sahoua health centers and the Taabo district hospital, as well as the manager of the Taabo pharmacy. Thanks to technical and administrative staff of Centre Suisse de Recherches Scientifique en Côte d'Ivoire, the Swiss Tropical and Public Health Institute (Swiss TPH) and the University Hospital of Heidelberg (Germany).

## Author contributions

**Conceptualization:** Tanoh Amany Serge Raymond N'Krumah, Bognan Valentin Koné, Yao Didier Koffi, Dognimin Ismael Coulibaly, Bassirou Bonfoh, Thomas Junghanss.

**Data curation:** Tanoh Amany Serge Raymond N'Krumah, Bognan Valentin Koné, Yao Didier Koffi, Dognimin Ismael Coulibaly, Abdoulaye Tall, Siaka Koné.

**Formal analysis:** Tanoh Amany Serge Raymond N'Krumah, Thomas Junghanss.

**Investigation:** Tanoh Amany Serge Raymond N'Krumah, Bognan Valentin Koné, Yao Didier Koffi, Dognimin Ismael Coulibaly, Siaka Koné.

**Methodology:** Tanoh Amany Serge Raymond N'Krumah, Bognan Valentin Koné, Yao Didier Koffi, Bassirou Bonfoh, Thomas Junghanss.

**Project administration:** Tanoh Amany Serge Raymond N'Krumah, Bognan Valentin Koné, Yao Didier Koffi.

**Resources:** Bassirou Bonfoh, Thomas Junghanss.

**Software:** Tanoh Amany Serge Raymond N'Krumah, Abdoulaye Tall.

**Supervision:** Bassirou Bonfoh, Thomas Junghanss.

**Validation:** Tanoh Amany Serge Raymond N'Krumah, Bognan Valentin Koné, Yao Didier Koffi, Dognimin Ismael Coulibaly, Abdoulaye Tall, Siaka Koné, Simone Toppino, Marija Stojkovic, Bassirou Bonfoh, Thomas Junghanss.

**Visualization:** Tanoh Amany Serge Raymond N'Krumah, Bassirou Bonfoh, Thomas Junghanss.

**Writing – original draft:** Tanoh Amany Serge Raymond N'Krumah, Bassirou Bonfoh, Thomas Junghanss.

**Writing – review & editing:** Tanoh Amany Serge Raymond N'Krumah, Bognan Valentin Koné, Yao Didier Koffi, Dognimin Ismael Coulibaly, Abdoulaye Tall, Siaka Koné, Simone Toppino, Marija Stojkovic, Bassirou Bonfoh, Thomas Junghanss.

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
