## [Decision Letter · Decision Letter 0]

31 Jan 2025

PNTD-D-25-00039

Health and Demographic Surveillance System (HDSS) based survey on burden, epidemiological and clinical characteristics of snakebite envenoming in Southern Côte d'Ivoire

Dear Dr. N'KRUMAH,

Thank you for submitting your manuscript to PLOS Neglected Tropical Diseases. After careful consideration, we feel that it has merit but does not fully meet PLOS Neglected Tropical Diseases's publication criteria as it currently stands. Therefore, we invite you to submit a revised version of the manuscript that addresses the points raised during the review process.

Please submit your revised manuscript within 60 days Apr 01 2025 11:59PM. If you will need more time than this to complete your revisions, please reply to this message or contact the journal office at plosntds@plos.org. Please include the following items when submitting your revised manuscript:

We look forward to receiving your revised manuscript.

Kind regards,

Wuelton Monteiro, Ph.D.

Section Editor

Wuelton Monteiro

Section Editor

Shaden Kamhawi

co-Editor-in-Chief

Paul Brindley

co-Editor-in-Chief

**Journal Requirements:**

3) Some material included in your submission may be copyrighted. According to PLOSu2019s copyright policy, authors who use figures or other material (e.g., graphics, clipart, maps) from another author or copyright holder must demonstrate or obtain permission to publish this material under the Creative Commons Attribution 4.0 International (CC BY 4.0) License used by PLOS journals. Please closely review the details of PLOSu2019s copyright requirements here: PLOS Licenses and Copyright. If you need to request permissions from a copyright holder, you may use PLOS's Copyright Content Permission form.

Potential Copyright Issues:

- Please confirm that you are the photographer of Figure 2, or provide written permission from the photographer to publish the photo(s) under our CC BY 4.0 license.

- Figure 1, Please provide a direct link to the base layer of the map (i.e., the country or region border shape) and ensure this is also included in the figure legend; and provide a link to the terms of use / license information for the base layer image or shapefile. We cannot publish proprietary or copyrighted maps (e.g. Google Maps, Mapquest) and the terms of use for your map base layer must be compatible with our CC BY 4.0 license.

4) Please ensure that the funders and grant numbers match between the Financial Disclosure field and the Funding Information tab in your submission form. Note that the funders must be provided in the same order in both places as well.

**Reviewers' Comments:**

Reviewer's Responses to Questions

**Key Review Criteria Required for Acceptance?**

**Methods**

-Are the objectives of the study clearly articulated with a clear testable hypothesis stated?

-Is the study design appropriate to address the stated objectives?

-Is the population clearly described and appropriate for the hypothesis being tested?

-Is the sample size sufficient to ensure adequate power to address the hypothesis being tested?

-Were correct statistical analysis used to support conclusions?

-Are there concerns about ethical or regulatory requirements being met?

Reviewer #1: The study clearly stated its objectives to estimate the burden and describe the epidemiological and clinical characteristics of snakebite envenoming based on population data. The study design appropriately addresses the study's goals, with the population clearly stated. The sample size power is sufficient, as 3,924 people were involved in the study, and the statistical analysis supports its conclusions. The study sought approval from the National Health and Life Sciences Ethics Committee.

Reviewer #2: The sampling technique does not seem at all clear or detailed.

I didn't see the sample size calculation anywhere. This has negative repercussions, as the small number of people who have experienced snakebite and venom does not provide adequate power to generalise the results.

In view of the small number of 11 people bitten by the snake over the study period, non-parametric statistics were indicated, but this is not the case for what the authors were able to do.

Can you justify the choice of design for this study ????

The article says nothing about how the sample size was calculated and nothing about the sampling technique used to select the household.

Reviewer #3: Small sample size but the study is well-designed and methodology explained very well.

Reviewer #4: - The study objectives are clearly presented.

- The population is clearly described and appropriate.

- The sample size is more than sufficient to ensure adequate power of the statistics generated.

- Correct statistical analysis has been used to support the conclusions.

**Results**

-Does the analysis presented match the analysis plan?

-Are the results clearly and completely presented?

-Are the figures (Tables, Images) of sufficient quality for clarity?

Reviewer #1: The results are presented well and clearly articulated.

Reviewer #2: The annual snakebite incidence per 100,000 population calculated from group

264 1 was 280.3 per 100,000 (CI: 140.0 - 501.0). The AHA snakebite case-fatality rate calculated

265 from group 2 was 2.0% (95% CI : 0.05 - 10.5) (Table 2). Can you explain these very wide confidence intervals?

What is the relevance of the choice of these two comparison groups?

The annual incidence

43 of snakebites was significantly higher in seasonal agricultural camps

44 (3296.7 cases per 100,000 inhabitants) than in villages (208.7 cases per 100,000 inhabitants).

45 inhabitants). Wouldn't this result be different if the size were larger?

Have you tried a non-parametric multivariate model with incidence as the dependent variable, given that univariate analyses are not reassuring?

45 inhabitants). Snakebites occur most often in the morning and at night. Have you used a statistical test?

Reviewer #3: Can't really assess if the analysis matches the analysis plan. The results are logically presented. The sample size justification is missing.

Reviewer #4: - Analysis corresponds to the analysis plan

- Results are presented clearly and completely

- Figures (tables, images) are of sufficient quality to ensure clarity

**Conclusions**

-Are the conclusions supported by the data presented?

-Are the limitations of analysis clearly described?

-Do the authors discuss how these data can be helpful to advance our understanding of the topic under study?

-Is public health relevance addressed?

Reviewer #1: The conclusion supports the data presented, outlines the limitations, and provides four recommendations that address public health relevance.

Reviewer #2: The authors do not explain how this data, which lacks power, can be used and generalised to advance knowledge.

Reviewer #3: The study effectively communicates its results making it easy for the reader to understand the burden of snakebites in the area. The authors have acknowledged the limitations.

Reviewer #4: - Conclusions are supported by the data presented

- The limitations of the analysis are clearly described

- The authors have clearly explained how the data can be useful in advancing overall understanding of the subject studied

- The subject poses a real public health problem

**Editorial and Data Presentation Modifications?**

Reviewer #1: None

Reviewer #2: (No Response)

Reviewer #3: Some minor revisions could be using consistent terminology for envenoming. There is some interchanging of envenomation and envenoming. Stick to one term. Also the discussion section needs a bit of reordering because it seems to jump about. Consider reordering for smoother flow.

Reviewer #4: Minor Revision

**Summary and General Comments**

Reviewer #1: The study is a valuable addition to the knowledge on snakebite research, an area that is silently affecting many rural individuals. Therefore, it is essential to determine its impact on our communities to mitigate it. The study highlights certain significant findings, such as the fact that most snakebite victims average 43 years of age, and first aid measures are often inadequately applied. These issues deserve attention, and I commend the authors for their contributions to this field of knowledge.

Reviewer #2: The authors need to improve the writing on the sampling technique and improve the power through a sample that allows us to generalise.

Parametric statistical tests are not appropriate given the small size of the subjects who suffered bites.

The univariate analyses alone do not seem to us to be at all robust for a good generalisation.

Reviewer #3: The study addresses a neglected public health issue using the HDDS data. It is good for the general public also in that it is highlighting treatment gaps such as the absence of antivenom at local health centers and the predominant reliance on traditional healers. There is an opportunity for a public health impact by targeting traditional healers as critical stakeholders in snakebite management.

Reviewer #4: Introduction

Line 87: Delete a space after in 2017

Lines 97-102: It is important here to specify whether these statistics concern the world or a given region.

Line 107: Please specify the scientific name of the viper species in question, in particular Echis carinatus.

Line 114 delete “- ‘ after ’national Household

Line 114-119: Here, you discuss national data for Côte d'Ivoire on snakebite incidence and mortality, based on a source from the 200th century, whereas more recent figures have been produced by studies by Habib et al. (2015). Why did you choose this option?

Line 147 study area

It's important here to better present or explain HDSS. Is HDSS a digital data collection platform or a research station? I read that “Le Système de Surveillance Sanitaire et Démographique de Taabo is a research station of the Centre Suisse de Recherches Scientifiques en Côte d'Ivoire (CSRS) located in south-central Côte d'Ivoire, about 150 km northwest of Abidjan.” If this is the case, I think that the title of the manuscript needs to be revised, in particular to read: “based on HDSS”. In any case, it is important to better explain the content of the HDSS.

Results

Line 311-313: Why did none of the victims receive antivenom when this is the main treatment recommended in hospitals? It would be important to mention the reason for this.

Line 314-315. It would be important to identify and mention the scientific names of the snakes responsible for the envenomations.

Discussion

Line 409: Delete “[”.

Line 410-411: Complete this sentence with : “and are therefore not overly involved in the rural activities that are the main source of snakebites.”

Line 455: Please specify the scientific name of the spitting cobra.

PLOS authors have the option to publish the peer review history of their article (what does this mean? ). If published, this will include your full peer review and any attached files.

**Do you want your identity to be public for this peer review?** For information about this choice, including consent withdrawal, please see our Privacy Policy .

Reviewer #1: No

Reviewer #2: No

Reviewer #3: **Yes: ** Naila Baig-Ansari

Reviewer #4: **Yes: ** DOSSOU Ayékotchami Jacques Calver, PhD

**Figure resubmission:**
---

## [Editor Report · Decision Letter 1]

13 Mar 2025

Dear Mr N'KRUMAH,

We are pleased to inform you that your manuscript 'Survey on the burden, epidemiological and clinical characteristics of snakebite envenoming in the Health Demographic Surveillance System (HDSS) of Taabo (Southern Côte d'Ivoire)' has been provisionally accepted for publication in PLOS Neglected Tropical Diseases.

Best regards,

Wuelton Monteiro, Ph.D.

Section Editor

Wuelton Monteiro

Section Editor

Shaden Kamhawi

co-Editor-in-Chief

Paul Brindley

co-Editor-in-Chief

---

## [Editor Report · Acceptance letter]

Dear Mr N'KRUMAH,

We are delighted to inform you that your manuscript, "Survey on the burden, epidemiological and clinical characteristics of snakebite envenoming in the Health Demographic Surveillance System (HDSS) of Taabo (Southern Côte d'Ivoire)," has been formally accepted for publication in PLOS Neglected Tropical Diseases.

Best regards,

Shaden Kamhawi

co-Editor-in-Chief

Paul Brindley

co-Editor-in-Chief
